# Assessment of the Anti-Thrombogenic Activity of Polyurethane Starch Composites

**DOI:** 10.3390/jfb13040184

**Published:** 2022-10-12

**Authors:** Jhoan F. Cespedes, Said Arévalo-Alquichire, Luis E. Diaz, Manuel F. Valero

**Affiliations:** 1Energy, Materials and Environmental Group, GEMA, Faculty of Engineering, Universidad de La Sabana, Chía 140013, Colombia; 2Master Program in Design and Management Process, Faculty of Engineering, Universidad de La Sabana, Chía 140013, Colombia; 3Department of Ophthalmology, Schepens Eye Research Institute of Mass Eye and Ear, Harvard Medical School, Boston, MA 02114, USA; 4Bioprospecting Research Group, GIBP, Faculty of Engineering, Universidad de La Sabana, Chía 140013, Colombia

**Keywords:** anti-thrombogenic property, polyurethanes composites, potato starch, zwitterionic starch, biomaterials, cardiovascular applications

## Abstract

The increasing morbidity and mortality of patients due to post-surgery complications of coronary artery bypass grafts (CABPG) are related to blood–material interactions. Thus, the characterization of the thrombogenicity of the biomaterial for cardiovascular devices is of particular interest. This research evaluated the anti-thrombogenic activity of polyurethanes–starch composites. We previously synthesized polyurethane matrices that were obtained from polycaprolactone diol (PCL), polyethylene glycol (PEG), pentaerythritol (PE), and isophorone diisocyanate (IPDI). In addition, potato starch (AL-N) and zwitterionic starch (AL-Z) were added as fillers. The anti-thrombogenic property was characterized by the clot formation time, platelet adhesion, protein absorption, TAT complex levels, and hemolysis. Additionally, we evaluated the cell viability of the endothelial and smooth muscle cells. Statically significant differences among the polyurethane matrices (P1, P2, and P3) were found for protein absorption and the blood clotting time without fillers. The polyurethanes composites with AL-Z presented an improvement in the anti-thrombogenic property. On the other hand, the composites with AL-Z reduced the viability of the endothelial cells and did not significantly affect the AoSCM (except for P1, which increased). These results classify these biomaterials as inert; therefore, they can be used for cardiovascular applications.

## 1. Introduction

Artificial, small diameter blood vessels (SDBVs) have been used as a solution for the scarcity of autologous grafts implemented in coronary artery bypass grafts (CABPGs) [1,2]. However, SDBVs have not yet mimicked the properties of autologous vessels (considered the gold standard for CABPGs), especially regarding their resistance to thrombus formation and biocompatibility in short- and long-term service [3]. This lack of blood compatibility in SDBVs has been reflected in complications during their implantation and function inside patients, such as thromboembolism, aneurysm, and cardiac tamponade [4].

It has been noted that biomaterials used for cardiovascular grafts face the hemostasis system, which refers to the mechanism that induces clot formation [5]. Therefore, when these devices disrupt this system, a blood clot is formed, causing the occlusion of the graft and post-CABPG complications [6]. One of the stages of the hemostasis mechanism related to biomaterials in cardiovascular applications is coagulation cascade activation. It has been proposed that blood contact materials activate clot formation by protein absorption and their attachment on the surface. In addition, they regulate the activation of the complement and immunology system, exacerbating the coagulation cascade and inflammation response [7,8]. The coagulation cascade is principally activated by interacting with the biomaterial surface, which is significant for the absorption and autoactivation of coagulation factor XII. In addition, it is raised by the adhesion and activation of platelets [9]. Furthermore, the activation of the complement and immunology systems has been connected to the inflammation response by the activation of neutrophils and macrophages that, in the presence of non-degradable implants, produce granulomas to phagocyte the biomaterial, inducing chronic inflammation [10]. Therefore, research efforts have focused on finding materials that reduce non-specific protein adsorption, platelet adhesion, and the activation of coagulation factors [11].

One of the most used polymers in cardiovascular applications is polyurethanes. This polymer is known for its biocompatibility, good mechanical properties, and easy processing in manufacturing technologies applied to cardiovascular devices [12]. However, the lack of knowledge about the thrombogenicity activity of polyurethanes has led to its study to improve its long-term anti-thrombotic response. Chemical backbone modifications [13,14,15], surface modifications [16,17,18], and the inclusion of fillers in the polymeric matrix [19,20] have been addressed. However, the characterization of composites with other assays different than hemolysis and cell viability is poorly reported. Additionally, zwitterionic moieties are characterized by their anti-fouling properties, which prevent the non-specific adsorption of proteins. Recently, the functionalization of polysaccharides with zwitterion has attracted attention in the field of biomaterials due to the combination of the biodegradability and natural resources of polysaccharides with the anti-fouling property of zwitterionic moieties [21]. Wang et al. [22] inserted the zwitterion moiety in starch using a Williamson etherification. They evaluated their resistance to protein absorption, finding that the functionalization of starch with the zwitterion significantly increased the anti-fouling property compared to poly(sulfobetaine methacrylate).

Our laboratory previously studied polyurethanes for cardiovascular applications, where they evaluated the influence of composition on the physic–mechanical and biological performance of PCL-PEG-PE polyurethanes [23,24,25]. Based on that work, we selected the compositions with the best biological [25] and biomechanical performance [23] to fabricate the composites presented in this work. Moreover, we characterized the chemical and mechanical properties of the polyurethane–starch composite [26]. These polyurethanes used potato starch and zwitterionic starch as fillers in three polyurethane compositions. We reported that adding potato starch increased the mechanical properties of the composites; moreover, adding zwitterionic starch improved the hydrophilicity of the polyurethane matrices with a major concentration of PEG. In addition, both fillers reduced the crystallinity of the polyurethane matrices.

Our review suggests that there is a lack of knowledge about the incorporation of zwitterionic starch as a filler in polyurethane matrices for cardiovascular applications. Therefore, this work evaluated the anti-thrombogenicity activity (according to ISO 10993-4) of polyurethane composites obtained from polycaprolactone diol (PCL), polyethylene glycol (PEG), pentaerythritol (PE), and isophorone diisocyanate (IPDI), adding potato starch (AL-N) and zwitterionic starch (AL-Z) as fillers. The results indicate that adding fillers, especially AL-Z, reduces clot formation related to the concentration of thrombin. In addition, these composite materials can be classified as inert due to the reduction in HUVEC viability.

## 2. Materials and Methods

### 2.1. Materials

Zwitterionic starch (AL-Z) was synthesized following the methodology reported in the previous article [26]. Polyethylene glycol (PEG, Mw~1000) was purchased from Merck KGaA (Darmstadt, Germany). Polycaprolactone diol (PCL-diol, Mw~2000), isophorone diisocyanate (IPDI), N, N-Dimethylformamide (anhydrous, 99.8%) (DMF), and potato starch (soluble) (AL-N) were purchased from Sigma-Aldrich (St. Louis, MO, USA). A Micro BCA™ Protein Assay Kit and Triton^®^ X-100 were obtained from Thermo Scientific (Waltham, MA, USA). A CytoTox 96^®^ Non-Radioactive Cytotoxicity Assay kit was provided by Promega (Woods Hollow Road, Madison, WI, USA) and a Human Thrombin-Antithrombin Complex ELISA Kit (TAT) by Abcam (Cambridge, UK). Sodium dodecyl sulfate (SDS), ethanol absolute, calcium chloride anhydrous (CaCl_2_), and bovine albumin serum (BSA) were purchased from Sigma-Aldrich (St. Louis, MO, USA). A 0.9% sodium chloride solution was acquired from Baxter (Deerfield, Illinois, USA). Phosphate buffer solution (PBS) was obtained from VWR^®^ (Radnor, PA, USA). Human umbilical vein endothelial cells (HUVECs), human aortic smooth muscle cells (AoSMC), an EGM BulletKit, and an SmGM-2 BulletKit were purchased from Lonza (Basel, Switzerland).

### 2.2. Synthesis of Polyurethane Composites

Polyurethane composites were obtained according to the methodology reported in a previous article [26]. First, a prepolymer was made from polycaprolactone diol (PCL), polyethylene glycol (PEG), and isophorone diisocyanate (IPDI) by melting the polyols at 110 °C and reacting them with IPDI at 70 °C and 300 rpm for 15 min. This reaction occurred in the presence of DMF. Second, the prepolymer solution was crosslinked by adding a solution of 0.497 M PE (the PE was dissolved in DMF at 110 °C) at 70 °C and 300 rpm for 15 min. The polyurethanes were obtained by maintaining a mol ratio of 1:1 NCO/mol OH. Third, filler (AL-N or AL-Z) was added to the reaction mixture and homogenized under vacuum at 70 °C for 15 min. Finally, the polyurethane composite solution was casted at 110 °C for 12 h. Three polyurethane matrix compositions were obtained by varying polyol concentration (in wt% and the crosslinking agent. In addition, the concentration of fillers was changed between 0% and 3 wt% (Table 1). For example, P1-1%-AL-N corresponds to the polyurethane matrix P1 composed of 5% PEG, 90% PCL, and 5% PE with 1 wt% of potato starch (AL-N).

The obtained polyurethane composites were confirmed by ATR-FTIR in a Cary 630 FTIR spectrometer (Agilent, Santa Clara, CA, USA). The spectra were recorded in a range from 650 cm^−1^ to 4000 cm^−1^ with a resolution of 2 cm^−1^. Additionally, the inclusion of the fillers was confirmed by scanning electron microscopy (SEM) with a TESCAN LYRA3 (Brno, Czech Republic) [27]. Figure 1 illustrates the synthesis process of polyurethane composites and the assays conducted to evaluate the anti-thrombogenicity activity.

### 2.3. Collection of Whole Human Blood

Fresh human whole blood was collected from healthy people aged 20–30 years in venous blood collection tubes with 3.2% buffered sodium citrate to make anti-coagulated blood. In addition, blood samples were obtained while maintaining aseptic conditions and guaranteed data confidentiality, according to the guidelines established by resolution No. 008430 de 1993 of the Ministry of Health of Colombia and the ethical committee of the Universidad de La Sabana. All subjects gave their informed consent for inclusion before participating in the study. The study was conducted in accordance with the Declaration of Helsinki, and the protocol was approved by the Ethics Committee of act 68 of 18 May 2018, project identification code ING-2018-2019.

### 2.4. Whole Blood Clotting Time

The clot formation time was evaluated according to the methodology proposed by Punnakitikashem et al. [28]. Briefly, samples of 6 mm diameter were sterilized with UV radiation for one h and stabilized in PBS for 24 h. In addition, recalcified blood (0.0091 M CaCl_2_) was obtained by mixing the anti-coagulated blood with a 0.1 M CaCl_2_ solution. Each stabilized sample in 2 mL microcentrifuge tubes was exposed to 50 µL of recalcified blood and incubated at ambient temperature (18–20 °C) for 10, 20, and 40 min. When exposure time finished, 1.5 mL of distilled water per sample was added and incubated for 5 min. Finally, an aliquot of 200 µL per sample was deposited in a 96-well plate to measure absorbance at 570 nm.

### 2.5. Hemolysis

The rupture of red cells was measured following the methods of Xu et al. [14]. First, samples of the composite materials (6 mm in diameter) were sterilized by UV radiation for one h and stabilized in PBS for 24 h. Then, a stabilized blood solution was made by adding 1.75 mL of anti-coagulated blood to 33.25 mL of a sterile 0.9% sodium chloride solution. The stabilized samples in 1.5 mL microcentrifuge tubes were exposed to stabilized blood (500 µL per sample) for two h at 37 °C and, when the exposure time ended, were centrifuged at 1000× *g* for 10 min. Finally, 100 µL of supernatant per sample was taken, and its absorbance (ABSs) was measured at 570 nm in a microplate reader. A positive control (ABSp) was made by adding 200 µL of anti-coagulated blood to 10 mL of distilled water. Negative control (ABSn) was made by blending 200 µL of anti-coagulated blood with 10 mL of a sterile 0.9% sodium chloride solution. The percentage of hemolysis was calculated using Equation (1).
% hemolysis = (ABSs − ABSn)/(ABSp − ABSn)(1)

### 2.6. Platelet Adhesion

Platelet adhesion was evaluated using the methodology proposed by Stahl et al. [29]. Platelet-rich plasma (PRP) was obtained by centrifuging the anti-coagulated blood at 120× *g* for 12 min. Samples with a diameter of 6 mm, sterilized by one h in UV radiation, and stabilized in PBS for 24 h, were exposed to PRP (200 µL per sample) at 37 °C for one h. Then, the samples were washed with PBS to remove any non-adhered platelet on the surface material, and the adhered platelets were lysed by soaking the samples in 1% triton (200 µL per sample) at 37 °C for 1 h. The LDH released by the adhered platelets was measured by a CytoTox 96^®^ Non-Radioactive Cytotoxicity Assay kit as per the manufacturer instructions.

### 2.7. Protein Absorption

The absorption of proteins was evaluated following the method of Jin et al. [30]. Briefly, samples of 6 mm in diameter, sterilized in UV radiation for 1 h and stabilized in PBS for 24 h, were incubated in a solution of bovine serum albumin (BSA) (0.1 µg/mL, 200 µL/sample) for 4 h. Then, each sample was washed three times with PBS and sonicated with 2% SDS solution (200 µL/sample) for 10 min to remove the absorbed proteins. Finally, the concentrations of the absorbed proteins by materials were determined by the protocol of the Micro BCA™ Protein Assay Kit.

### 2.8. Thrombin–Anti-Thrombin (TAT) Complex

Samples (previously sterilized in UV for 1 h and stabilized in PBS for 24 h) located in 2 mL microcentrifuge tubes were exposed to anti-coagulated blood (1 mL per sample) at 37 °C for 1 h and centrifuged at 3000× *g* for 10 min. The concentration of TAT complex in the supernatant was determined according to the protocol of the Human Thrombin-Anti-thrombin (TAT) Complex ELISA Kit [28].

### 2.9. HUVECs and AoSMC Cultures

Endothelial and smooth muscle cells were cultured in 75 cm^2^ flasks at 37 °C with 5% CO_2_ for 8 passages. The medium for cells was changed every 2 days until the flask reached 90% confluence, at which point, a passage was carried out. The cells were plated at passage 9 for assays. EGM and SmGM-2 were used to grow the HUVECs and AoSMCs, respectively.

### 2.10. Cell Viability

Samples of the composite materials of 6 mm in diameter (sterilized in UV radiation for 1 h and stabilized in SmGM-2 culture medium for 24 h) were placed in the wells of a 96-well plate with an AoSMC suspension at a density of 4 × 10^5^ cell per mL and incubated at 37 °C with 5% CO_2_ for 24 h. An aliquot of 50 µL per sample was taken and deposited in another 96-well plate to measure the LDH released following the protocol of the CytoTox 96^®^ Non-Radioactive Cytotoxicity Assay kit [25]. Cell viability of HUVECs was measured following the same protocol.

The percentage cell viability was calculated using Equation (2).
% cell viability = 1 − (ABSs − ABSn)/(ABSp − ABSn)(2)

### 2.11. Statistics Analysis

Three independent samples were tested by assay. The data were analyzed using two-way ANOVA, considering the composition of the matrix (factor 1) and the concentration of the filler (factor 2). In addition, the polyurethane matrices without fillers were analyzed using one-way ANOVA. The samples were compared by a post hoc Tukey comparison test. Statical analysis was conducted using RStudio software version 1.3.1093.

## 3. Results and Discussion

The solid polyurethane composites used in this research were the same as those obtained in a previous study [26]. The materials were characterized in terms of their physicochemical, mechanical, and thermal properties. We reported the ^1^H NMR spectrum of AL-Z; by ATR-FTIR (Appendix A), we demonstrated a complete reaction of isocyanate due to the absence of a 2225 cm^−1^ band. The inclusion of the fillers in the polyurethane matrices were confirmed by SEM (Appendix A) and EDS (Appendix A). Additionally, XRD spectra showed that including fillers reduced the degree of crystallinity of the polyurethane matrices. The contact angle of P3 with AL-Z exhibited a significant reduction compared with P3 without fillers. These results indicate that the inclusion of AL-Z improves the hydrophilicity in the polyurethane matrices with a major concentration of PEG as P3 and the hydrophobicity for matrices with the largest concentration of PCL. In this research, the polyurethane composites were characterized according to their thromboresistance property.

To evaluate the formation of clots induced by the polyurethane composites, a blood clotting time test was conducted. The results for the polyurethane composites without fillers and those with AL-N and AL-Z are shown in Table 2. These results correspond to the absorbance of hemoglobin released by the lysed red blood cells that were not involved in clot formation. Therefore, at high absorbance values, fewer clots developed [31]. As can be seen in Table 2, polyurethane matrix P3 without fillers behaved most similarly to the control (recalcified blood alone) compared to P2 and P1. Also, P3-0%-AL showed less formation of a clot at 20 min to a significant degree than P1-0%-AL and P2-0%-AL. This better resistance to the clot formation of P3 in comparison to the P1 and P2 matrices could be attributed to the major concentration of PEG, which is reported as an anti-fouling polymer [32].

The addition of AL-Z filler showed a significant improvement in the resistance to clot formation (Table 2) for P2-2%-AL-Z compared to P2-0%-AL and P2 with AL-N as a filler at 20 min in the assay. A similar significant effect was identified for P3-2%-AL-Z compared to P3-0%-AL and P3 composites with AL-N at 20 min and 40 min. These results could be a result of the increase in the hydrophilicity of the polyurethane matrix, results reported in the previous article [26], associating this behavior to the reduction in the crystallinity of PEG, which improved its chain mobility at the surface [33] and allowed for the development of a tiny layer of water, which is related to the prevention of the protein absorption and activation of the coagulation cascade [32].

The thrombogenicity characterization of the polyurethane composites was conducted according to ISO 10993-4 to establish the correlation between the synthesized materials and the thrombus formation. Among the in vitro tests proposed by this standard and carried out in this work, the hemolysis, platelet adhesion, and TAT complex formation (concentration of thrombin) were evaluated. Additionally, protein absorption was studied due to its correlation with the initial stage of clot formation [34].

The results of the hemolysis assay are shown in Table 3 for polyurethane composites without fillers and composites with AL-N and AL-Z. All materials exhibited a value of less than 2% of hemolysis; therefore, these materials are classified as non-hemolytic according to the ISO 10993-4. Biomaterials must ensure a non-hemolytic property due to red blood cells lysing after their contact with the surface of the material; this could elicit the coagulation cascade by the release of hemoglobin, which can reduce NO, a compound that activates platelets. In addition, hemoglobin can induce the expression of adhesion molecules (ICAM-1 and VCAM-1) presented in endothelial cells, promoting the obstruction of the blood vessel and, consequently, cell damage [35].

The activation and adhesion of platelets is the assay most used to characterize the anti-thrombotic property of biomaterials due to its direct correlation with clot formation. Platelet activation induces platelet aggregation, which forms a platform that contains phospholipids (compounds present in the structure of platelets) related to the activation of factor X from the coagulation cascade and causing the activation of other factors that results in the production of fibrin (the principal polymer of clots) [9]. Platelet adhesion on the polyurethane composites was evaluated by the release of LDH from lysed platelets adhered to the surfaces of the composites. The results show that polyurethane matrices without fillers and composites with AL-N or AL-Z (Table 3) correspond to the absorbance of LDH at 490 nm; therefore, higher absorbance values are associated with major platelet adhesion. In Table 3, it can be seen that polyurethane matrices without fillers had no significant effect on biomaterial from the GORE^®^ PROPATEN^®^ vascular graft, which contains heparin at the surface, a compound characterized as an anti-thrombotic agent [36]. Furthermore, adding fillers (AL-N or AL-Z) did not exhibit significant effects (Table 3). These results are consistent with the literature reporting on materials that contain PEG, which attribute the resistance to platelet adhesion to the water layer bound around the hydrophilic groups of this polymer [37,38].

Protein absorption in biomaterials (especially fibrinogen) is associated with the activation of the intrinsic pathway of the coagulation cascade, complement system, and immunology response (the activation of neutrophils and macrophages) [8]. However, some investigations [39,40,41] have reported that the selective adsorption of albumin, the major protein in human blood plasma [42], reduces the adhesion of platelets. This is because albumin lacks an amino acid sequence that binds with the activation receptor of platelets [34]. Table 3 shows that the protein absorption in the polyurethane matrices P2 and P3 (without fillers) is significantly larger than in P1. This phenomenon could be driven by the electrostatic interactions between the PEG segment and albumin. Bremmell et al. [39], who studied the electrostatic effect of albumin adsorption on a modified surface of silicon with PEG-CHO and PEG-SO_3_, found the interaction between the protein and surface material depends on these electrostatic interactions. Including a filler (AL-N or AL-Z) in polyurethane matrices did not significantly affect protein adsorption (Table 3).

The formation of thrombin is crucial in the coagulation cascade because this enzyme converts the fibrinogen to fibrin (a polymer that compounds the clot) [9]. Therefore, the thrombin concentration was evaluated in this study using a TAT complex assay. In Table 3, it can be seen that the polyurethane matrices without fillers significantly increased the concentration of thrombin compared to the control (anti-coagulated fresh blood alone). In Table 3, it can be seen that the addition of fillers significantly reduced the concentration of thrombin. The inclusion of AL-Z at the highest concentration (3% wt) especially reduced the TAT complex until it was similar to the control. These results could be attributed to the increase in the mobility chains of soft segments, mainly the PEG segment, due to the reduction in the soft domain crystallinity, as was reported in the previous article [26]. This reduction in crystallinity caused by the addition of AL-Z allows for better hydrophilicity of the surface and the formation of a water barrier that could reduce the activation of factor XII.

The biocompatibility of the polyurethane composites was evaluated by exposing HUVECs and AoSMC to composites and assessing the cell viability. It is well known that the proliferation of adherent cells depends on their attachment to a surface; otherwise, cells can suffer apoptosis [43]. As seen in Table 4, the polyurethane matrices without fillers did not exhibit any significant difference in the viability of the cells. Moreover, adding fillers (AL-N or AL-Z) reduced the viability of the HUVECs for the P2 and P3 polyurethane composites to a significant degree compared to their respective polyurethane matrices without fillers. This reduction was larger when AL-Z was used as a filler. These results could be attributed to an increase in the hydrophilicity of the surfaces, which reduces the cell adhesion and, therefore, its capacity to proliferate [22]. Likewise, the inclusion of AL-Z in the P1 polyurethane matrix increased the cell viability of AoSMC. The surface of P1 was more hydrophobic (contact angle between 103.7–110.3°) compared with the other matrices (99.5–39.4°). This suggested that the surface requirements for HUVECs and AoSMCs adhesion are different, and it is of interest in the design of cardiovascular applications.

From these results, it could be stated that the addition of AL-Z improved certain characteristics of the polyurethane matrices. The materials studied here postulate as candidates for cardiovascular applications. However, further studies are required, such as in vivo models, to validate the presented results.

## 4. Conclusions

The anti-thrombogenic activity of the materials was characterized by the blood clotting time. The correlations between the materials and thrombus formation were studied according to ISO 10993-4, evaluating the hemolysis, platelet adhesion, and TAT complex. These assays were complemented by developing a protein adsorption assay and a HUVECs and AoSMC viability assay. The blood clotting time assay showed that polyurethane matrix P3 without fillers had a better thromboresistance property than P1 and P2. In addition, the hemolysis assay showed that all of the materials were not hemolytic. According to the platelet adhesion assay, all the materials showed similar behavior compared to the heparin-modified commercial grafts. The protein absorption results showed that the polyurethane matrices P2 and P3 had albumin adsorption; however, it is necessary to evaluate the Vroman effect with a blending of proteins to establish the behavior of the protein absorption of the materials. According to the TAT complex assay, AL-Z improved the resistance to thrombus formation due to the reduction in the thrombin concentration. It must be stated that the inclusion of AL-Z in the polyurethane matrices improved certain characteristics and are candidates for cardiovascular applications. Additionally, biomaterials should be characterized with in vivo assays to establish if they can be applied in that field.

## Figures and Tables

**Figure 1 jfb-13-00184-f001:**
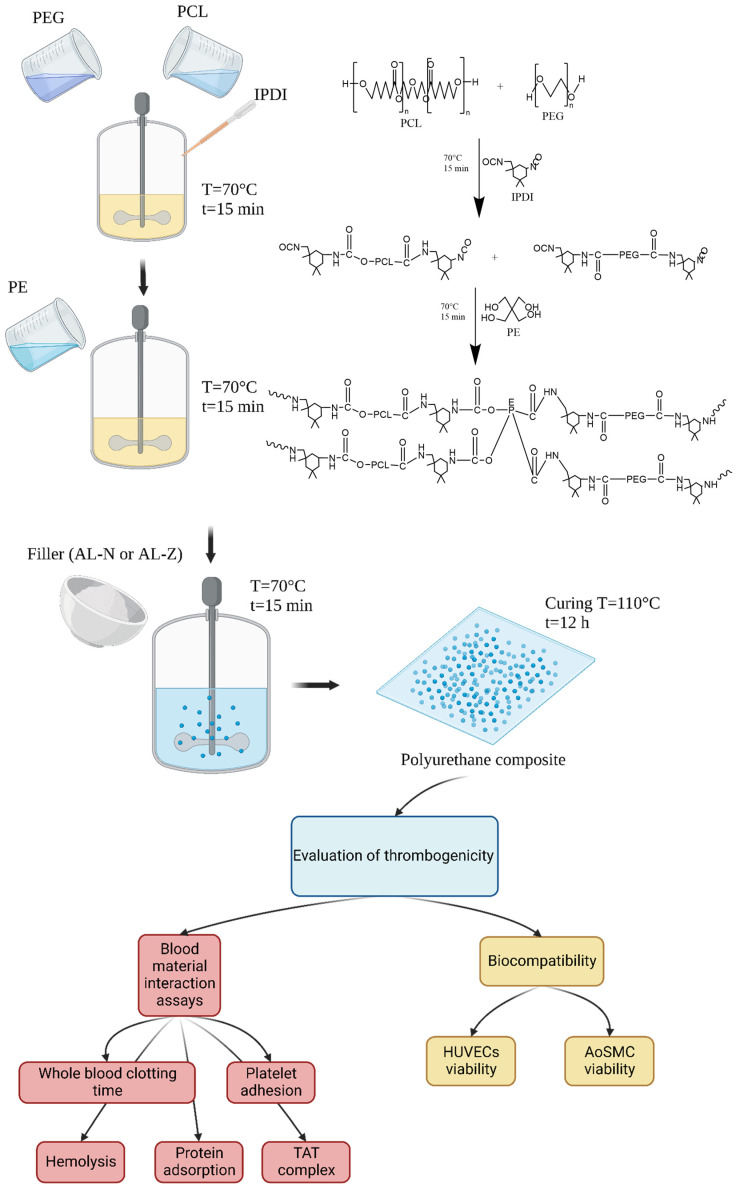
Synthesis of composites and assays conducted in this study. This figure was created with BioRender.

**Table 1 jfb-13-00184-t001:** Composition of polyurethanes composites.

PEG *	PCL *	PE *	PU-Matrix	Starch	Sample **
5%	90%	5%	P1	0%	P1-0%-AL
				1%	P1-1%-AL-(N or Z)
				2%	P1-2%-AL-(N or Z)
				3%	P1-3%-AL-(N or Z)
45%	45%	10%	P2	0%	P2-0%-AL
				1%	P2-1%-AL-(N or Z)
				2%	P2-2%-AL-(N or Z)
				3%	P2-3%-AL-(N or Z)
46.25%	46.25%	7.5%	P3	0%	P3-0%-AL
				1%	P3-1%-AL-(N or Z)
				2%	P3-2%-AL-(N or Z)
				3%	P3-3%-AL-(N or Z)

* Percentage of polyols are in wt. ** AL alludes to starch (natives or zwitterionic). N relates to native and Z to zwitterionic.

**Table 2 jfb-13-00184-t002:** Clot formation time of recalcified blood in contact with polyurethane composites.

PU	Starch	Concentration	Absorbance at 570 nm **		
			t = 10 min	t = 20 min	t = 40 min
Control			0.32 ± 0.04 ^a^	0.39 ± 0.11 ^a^	0.11 ± 0.04 ^a^
P1	AL-0%		0.09 ± 0.03 ^a^	0.08 ± 0.09 ^a^	0.06 ± 0.02 ^a^
	AL-N	1%	0.13 ± 0.05 ^a^	0.09 ± 0.06 ^a^	0.07 ± 0.04 ^a^
		2%	0.18 ± 0.10 ^a^	0.08 ± 0.06 ^a^	0.12 ± 0.06 ^a^
		3%	0.15 ± 0.09 ^a^	0.06 ± 0.05 ^a^	0.04 ± 0.01 ^a^
	AL-Z	1%	0.24 ± 0.09 ^a^	0.12 ± 0.05 ^a^	0.07 ± 0.02 ^a^
		2%	0.29 ± 0.15 ^a^	0.07 ± 0.05 ^a^	0.12 ± 0.05 ^a^
		3%	0.20 ± 0.05 ^a^	0.14 ± 0.08 ^a^	0.11 ± 0.06 ^a^
	*p*-values				
	Starch		0.072	0.450	0.277
	Concentration		<0.001 *	0.001 *	0.047
	Starch/Concentration		0.624	0.829	0.417
Control			0.32 ± 0.04 ^a^	0.39 ± 0.11 ^ace^	0.11 ± 0.04 ^abc^
P2	AL-0%		0.22 ± 0.16 ^a^	0.08 ± 0.05 ^be^	0.08 ± 0.07 ^ac^
	AL-N	1%	0.14 ± 0.04 ^a^	0.08 ± 0.03 ^bce^	0.04 ± 0.01 ^a^
		2%	0.27 ± 0.08 ^a^	0.05 ± 0.02 ^be^	0.06 ± 0.01 ^ab^
		3%	0.19 ± 0.09 ^a^	0.10 ± 0.06 ^bce^	0.08 ± 0.02 ^abc^
	AL-Z	1%	0.24 ± 0.09 ^a^	0.07 ± 0.02 ^be^	0.05 ± 0.01 ^a^
		2%	0.33 ± 0.15 ^a^	0.31 ± 0.17 ^ce^	0.26 ± 0.06 ^bc^
		3%	0.23 ± 0.14 ^a^	0.15 ± 0.04 ^e^	0.23 ± 0.06 ^c^
	*p*-values				
	Starch		0.362	0.022 *	0.004 *
	Concentration		0.207	<0.001 *	0.002 *
	Starch/Concentration		0.832	0.011 *	0.036 *
Control			0.32 ± 0.04 ^a^	0.39 ± 0.11 ^a^	0.11 ± 0.04 ^abcd^
P3	AL-0%		0.19 ± 0.08 ^a^	0.45 ± 0.17 ^a^	0.08 ± 0.04 ^acd^
	AL-N	1%	0.25 ± 0.11 ^a^	0.33 ± 0.14 ^a^	0.10 ± 0.03 ^abcd^
		2%	0.16 ± 0.057 ^a^	0.31 ± 0.16 ^a^	0.09 ± 0.06 ^acd^
		3%	0.16 ± 0.07 ^a^	0.20 ± 0.04 ^a^	0.12 ± 0.04 ^abcd^
	AL-Z	1%	0.24 ± 0.07 ^a^	0.22 ± 0.12 ^a^	0.28 ± 0.08 ^bd^
		2%	0.27 ± 0.01 ^a^	0.20 ± 0.07 ^a^	0.21 ± 0.04 ^cd^
		3%	0.28 ± 0.03 ^a^	0.23 ± 0.02 ^a^	0.19 ± 0.04 ^d^
	*p*-values				
	Starch		0.064	0.407	0.005 *
	Concentration		0.091	0.019 *	0.038 *
	Starch/Concentration		0.231	0.676	0.122

** Samples with the same letter do not have a statistically significant difference. Samples with a different letters have significant differences. * Effects are statistically significant (*p*-value < 0.05).

**Table 3 jfb-13-00184-t003:** Blood–composites contact assays.

PU	Starch	Concentration	Hemolysis (%) **	Platelet Adhesion (Absorbance) **	Protein Absorption (µg/mL) **	TAT Complex (ng/mL) **
Control			-	0.31 ± 0.05 ^a^	-	1.15 ± 0.28 ^acde^
P1	AL-0%		0.84 ± 1.01 ^a^	0.39 ± 0.04 ^a^	0.00	2.71 ± 0.34 ^b^
	AL-N	1%	0.35 ± 0.32 ^a^	0.42 ± 0.17 ^a^	0.00	3.12 ± 0.38 ^b^
		2%	0.47 ± 0.60 ^a^	0.38 ± 0.04 ^a^	0.00	3.13 ± 0.38 ^b^
		3%	1.02 ± 0.36 ^a^	0.38 ± 0.05 ^a^	0.00	2.99 ± 0.25 ^b^
	AL-Z	1%	0.32 ± 0.56 ^a^	0.37 ± 0.09 ^a^	0.00	1.81 ± 0.26 ^c^
		2%	0.18 ± 0.15 ^a^	0.32 ± 0.04 ^a^	0.00	1.66 ± 0.25 ^d^
		3%	0.38 ± 0.66 ^a^	0.37 ± 0.04 ^a^	0.00	1.55 ± 0.19 ^e^
	*p*-values					
	Starch		0.379	0.379	0.573	<0.001 *
	Concentration		0.425	0.166	0.998	<0.001 *
	Starch/Concentration		0.815	0.908	0.680	<0.001 *
Control			-	0.31 ± 0.05 ^a^	-	1.15 ± 0.28 ^ac^
P2	AL-0%		0.58 ± 0.67 ^a^	0.37 ± 0.06 ^a^	11.46 ± 1.20 ^ab^	3.40 ± 0.23 ^b^
	AL-N	1%	0.20 ± 0.35 ^a^	0.38 ± 0.07 ^a^	13.44 ± 1.10 ^ab^	2.00 ± 1.13 ^c^
		2%	0.14 ± 0.25 ^a^	0.40 ± 0.05 ^a^	12.76 ± 1.91 ^ab^	1.35 ± 0.28 ^c^
		3%	0.76 ± 0.66 ^a^	0.49 ± 0.12 ^a^	15.55 ± 2.78 ^a^	1.19 ± 0.12 ^c^
	AL-Z	1%	0.20 ± 0.35 ^a^	0.36 ± 0.06 ^a^	11.89 ± 2.90 ^ab^	1.36 ± 0.40 ^c^
		2%	0.55 ± 0.96 ^a^	0.34 ± 0.05 ^a^	7.86 ± 2.46 ^b^	1.28 ± 0.26 ^c^
		3%	0.32 ± 0.56 ^a^	0.31 ± 0.04 ^a^	12.95 ± 3.71 ^ab^	1.21 ± 0.21 ^c^
	*p*-values					
	Starch		0.977	0.035 *	0.031 *	0.437
	Concentration		0.677	0.242	0.055	<0.001 *
	Starch/Concentration		0.688	0.124	0.356	0.757
Control			-	0.31 ± 0.05 ^ab^	-	1.15 ± 0.28 ^ac^
P3	AL-0%		0.55 ± 0.20 ^a^	0.41 ± 0.05 ^ab^	9.08 ± 1.06 ^a^	2.81 ± 0.30 ^b^
	AL-N	1%	0.32 ± 0.20 ^a^	0.40 ± 0.06 ^ab^	10.92 ± 2.41 ^a^	1.58 ± 0.89 ^c^
		2%	0.79 ± 0.18 ^a^	0.44 ± 0.06 ^ab^	13.87 ± 2.89 ^a^	1.04 ± 0.02 ^c^
		3%	0.26 ± 0.30 ^a^	0.48 ± 0.06 ^a^	15.17 ± 3.47 ^a^	1.03 ± 0.08 ^c^
	AL-Z	1%	0.57 ± 0.27 ^a^	0.31 ± 0.06 ^ab^	12.43 ± 1.32 ^a^	1.16 ± 0.02 ^c^
		2%	0.35 ± 0.53 ^a^	0.30 ± 0.05 ^b^	11.46 ± 5.94 ^a^	1.10 ± 0.24 ^c^
		3%	0.57 ± 0.36 ^a^	0.32 ± 0.05 ^ab^	13.14 ± 4.31 ^a^	1.09 ± 0.11 ^c^
	*p*-values					
	Starch		0.862	<0.001 *	0.588	0.755
	Concentration		0.734	0.048 *	0.088	<0.001
	Starch/Concentration		0.188	0.054	0.701	0.844

** Samples with the same letter are not statistically significantly different. Samples with different letters are significantly different. * Effects are statistically significant (*p*-value < 0.05).

**Table 4 jfb-13-00184-t004:** HUVECs and AoSMC viability in the presence of polyurethane composites.

PU	Starch	Concentration	HUVECs Viability (%) **	AoSMC Viability (%) **
Control			100%	100%
P1	AL-0%		82.23 ± 0.96 ^a^	106.59 ± 17.02 ^a^
	AL-N	1%	88.28 ± 4.57 ^a^	99.73 ± 4.15 ^a^
		2%	89.74 ± 2.74 ^a^	106.73 ± 3.08 ^a^
		3%	92.49 ± 3.09 ^a^	105.14 ± 4.72 ^a^
	AL-Z	1%	85.98 ± 4.00 ^a^	195.23 ± 22.38 ^b^
		2%	83. 79 ± 9.29 ^a^	172.40 ± 37.82 ^b^
		3%	88.73 ± 2.48 ^a^	195.10 ± 14.20 ^b^
	*p*-values			
	Starch		0.107	<0.001 *
	Concentration		0.031 *	0.003 *
	Starch/Concentration		0.683	0.001 *
Control			100%	100%
P2	AL-0%		83.24 ± 3.96 ^a^	100.26 ± 2.39 ^a^
	AL-N	1%	71.70 ± 21.26 ^ab^	105.80 ± 5.38 ^a^
		2%	79.94 ± 11.29 ^a^	98.65 ± 16.79 ^a^
		3%	64.74 ± 12.67 ^ab^	83.72 ± 33.96 ^a^
	AL-Z	1%	37.91 ± 10.28 ^b^	169.49 ± 58.56 ^a^
		2%	33.70 ± 11.82 ^b^	141.18 ± 18.07 ^a^
		3%	34.89 ± 22.37 ^b^	166.05 ± 46.96 ^a^
	*p*-values			
	Starch		<0.001 *	0.002 *
	Concentration		0.003 *	0.239
	Starch/Concentration		0.059	0.152
Control			100%	100%
P3	AL-0%		77.01 ± 9.35 ^a^	81.79 ± 20.28 ^a^
	AL-N	1%	77.93 ± 5.21 ^a^	78.13 ± 9.05 ^a^
		2%	69.78 ± 11.13 ^ac^	75.03 ± 15.91 ^a^
		3%	79.12 ± 6.19 ^a^	71.68 ± 36.78 ^a^
	AL-Z	1%	35.44 ± 6.71 ^bc^	79.06 ± 43.76 ^a^
		2%	40.11 ± 14.60 ^bc^	115.04 ± 54.41 ^a^
		3%	45.78 ± 7.41 ^c^	112.08 ± 22.78 ^a^
	*p*-values			
	Starch		<0.001 *	0.133
	Concentration		0.003 *	0.771
	Starch/Concentration		0.006 *	0.512

** Samples with the same letter are not statistically significantly different. However, samples with different letters have significant differences. * Effects are statistically significant (*p*-value < 0.05).

## Data Availability

Data can be found in Appendix A, where spectra and images related to materials are available.

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
