# Peer review of "Assessment of the Anti-Thrombogenic Activity of Polyurethane Starch Composites"

_jfb, 2022, doi:10.3390/jfb13040184_

Round 1

Reviewer 1 Report

The manuscript entitled, "Assessment of the anti-thrombogenic activity of polyurethane starch composites" is required following revision.

Comments:

1. Photograph and SEM morphology of polyurethane-starch composite should be incorporated.

2. What about the coagulation cascade of the prepared composities.

3.  What about RBC/Platelet aggregation property of the prepared composities.

Author Response

Reviewer 1
RE: Changes suggested by reviewer 1
Dear reviewer 1:
We appreciate your comments to improve this paper. Therefore, we made the following change to the document:

Reviewer 2 Report

This is a well-written, comprehensive paper. In my opinion, this paper is a welcome addition to the current literature concerning the development and formulation of inert biomaterials that can be used for cardiovascular applications.

The idea of this manuscript is good, and the study is technically well performed. However, I have minor concerns and comments, which are outlined below:

2.2. Synthesis of polyurethane composites

-          Authors should identify if the % is in wt or vol.

-          In the text, the explanation of the formula P1-1%-AL-N as an example is unclear. Why 10% PCL? The authors should probably write 90%!

Table 1

-          Authors should identify if the % is in wt or vol.

-          Authors should change "ó" to “or”.

2.3. Collection of human whole blood

-          Authors should identify the average age of donors (if it’s available).

3. Results and discussion

-          Regarding M&M section, figures of some ATR-FTIR spectra and SEM images should be presented in the manuscript.

-          The “Error! Reference source not found.” should be corrected.

-          For a better visualisation, Table 4 (i.e., HUVECs and AoSMC viability in the presence of polyurethane composites) can be converted into figures (histograms of viability as a function of condition).

-          Some chemical interpretations and discussions should be presented according to the effect of filler type (potato starch and zwitterionic starch).

Author Response

Reviewer 2
RE: Changes suggested by reviewer 2
Dear reviewer 2:
We appreciate your comments to improve this paper. Therefore, we made the following change to the document:

Reviewer 3 Report

Dear Editor, in the submitted paper polyurethane matrices have been prepared polycaprolactone diol (PCL), polyethylene glycol (PEG), pentaerythritol (PE), and isophorone diisocyanate (IPDI). To these matrices potato starch (AL-N) or zwitterionic starch (AL-Z) were added as fillers. The anti-thrombogenic property was characterized by the clot formation time, platelet adhesion, protein absorption, TAT complex levels, and hemolysis.

I have seen the paper and I must say that I have many concerns for the used materials. Isophorone diisocyanate is a toxic material while the addition of PCL and PEG is not explained. Why are they used? PCL will give a stiffer material while PEG a softer. What is the aim of this study? I really disagree with authors opinion that the addition of PCL will increase segment mobility chains.

The prepared polyurethanes have not characterized concerning their properties and even if the reaction was proceeded. FTIR and NMR studies are needed. What is the extent of monomer reaction and unreacted monomers? This very crucial since, as I said, some of the used monomers are toxic.

What is the physical state of these composites and their stability? These are not reported.

It is not clear what starch will enhance with its addition to PU matrices. What is its benefit compared with other polysaccharides like chitosan, which extensively used for such applications, or to cellulose? Furthermore, which are the benefits of starch use? As was reported the addition of fillers (AL-N or AL-Z)
reduced the viability of the HUVECs!!! Furthermore, it was reported that the blood clotting time assay showed that polyurethane matrix P3 without fillers had a better thromboresistance property in comparison to P1 and P2.

So, the advantages of the used fillers are questionable!!!

Taking into account all above mentioned I do not feel that this paper has any novelty concerning its findings and used materials. There are lot of scientific concerns and the prepared composites are not at all characterized .For all these I propose rejection.  

Author Response

Reviewer 3
RE: Changes suggested by reviewer 3
Dear reviewer 3:
We appreciate your comments to improve this paper. Therefore, we made the following change to the document:

Round 2

Reviewer 1 Report

Accept

Author Response

Dear reviewer 1:
We appreciate your comments to improve this paper. 

Reviewer 3 Report

Dear Editor, I have seen the authors’ responses on my comments and it is clear that the most of them cannot be properly addressed. Especially the toxicity of the used isocyanates and the benefits of the used starch. For this reason, I propose to reject it again for publication. It is just one paper without any novelty and possibility these materials to find any application.

Author Response

Dear reviewer 3:

Under your comments to the paper, we made the following responses:
